# Superior Electrocatalytic Activity of MoS_2_-Graphene as Superlattice

**DOI:** 10.3390/nano10050839

**Published:** 2020-04-27

**Authors:** Alejandra Rendón-Patiño, Antonio Domenech-Carbó, Ana Primo, Hermenegildo García

**Affiliations:** 1Instituto de Tecnología Química (CSIC-UPV) and Department of Chemistry, Consejo Superior de Investigaciones Científicas-Universitat Politècnica de Valencia, Avenida de los Naranjos s/n, 46022 Valencia, Spain; alrenpa@itq.upv.es; 2Departamento de Química Analítica, Universitat de Valencia, Av. Del Dr. Moliner s/n, 46100 Burjassot, Spain; antonio.domenech@uv.es

**Keywords:** superlattice, 2d materials, electrocatalytic

## Abstract

Evidence by selected area diffraction patterns shows the successful preparation of large area (cm × cm) MoS_2_/graphene heterojunctions in coincidence of the MoS_2_ and graphene hexagons (superlattice). The electrodes of MoS_2_/graphene in superlattice configuration show improved catalytic activity for H_2_ and O_2_ evolution with smaller overpotential of +0.34 V for the overall water splitting when compared with analogous MoS_2_/graphene heterojunction with random stacking.

## 1. Introduction

There is considerable interest in developing noble metal-free electrodes that could efficiently perform water splitting due to the current change from fossil fuels to renewable electricity [1,2,3,4]. In this context, it has been reported that MoS_2_ could be an efficient electrocatalyst for the oxygen evolution reaction (OER) replacing IrO_2_ and Pt [5,6]. MoS_2_ also shows electrocatalytic activity for hydrogen evolution reaction (HER) [7,8,9] and, therefore, it could be an ideal material to perform both redox processes of overall water splitting.

Regarding the use of MoS_2_ as electrocatalyst, it has been shown that the assembly of this chalcogenide with graphene improves its performance as electrodes [10,11,12]. Graphene introduces electrical conductivity favouring electron transfer from the catalytic site on MoS_2_ to the external circuit. For this reason, there has been considerable interest in developing different procedures for the preparation of MoS_2_/graphene heterojunctions to be used in electrolysis [13,14,15] 

In this context, a procedure consisting in the one-step formation of MoS_2_ strongly interacting with graphene has been recently reported. The process is based on the pyrolysis of (NH_4_)_2_MoS_4_ and chitosan as precursors of MoS_2_ and N-doped defective graphene, respectively [16]. One of the advantages of this procedure was that chitosan allows for the formation of nanometric films on arbitrary, non-conductive substrates, such as quartz or ceramics, which, after transformation on (N)G, becomes electrically conductive and can be used as electrode [17,18]. Also related to the present study is a recent report from us on the formation of large area boron nitride/graphene heterojunctions with superlattice configuration [19].

Since both boron nitride and graphene are two-dimensional (2D) materials with very similar lattice parameters, the term superlattice refers to the coincidence of the hexagons of boron nitride overlapping those of graphene. Fundamental studies have shown that superlattice configuration of 2D materials heterojunctions can give an assembly with tunable electrical conductivity in contrast to the random heterojunctions of the two materials [20,21]. This modulation of the electron mobility through the graphene layer by the underlying boron nitride could be reflected in some unique properties of the superlattice heterojunction, particularly those that are related to electrochemistry and electrocatalysis [22,23,24]. In view of these precedents, it would be of interest to determine the electrocatalytic behaviour of MoS_2_/graphene heterojunction in superlattice configuration, particularly when considering the well-known activity of MoS_2_ for important reactions, such as HER, OER, and oxygen reduction reaction (ORR) processes [5].

This manuscript reports a novel procedure for the preparation of large area (cm × cm) films of MoS_2_/graphene heterojunction in superlattice configuration. It will be shown that this fl-MoS_2_/graphene material (fl meaning few layers) exhibits much lower onset potential for HER and OER than an analogous electrode prepared with random MoS_2_/G heterojunction, illustrating the advantages of lattice matching to improve the electrochemical performance.

## 2. Materials and Methods

### 2.1. Methods-Exfoliation MoS_2_ and Preparation of MoS_2_/fl-G

Molybdenum sulfide exfoliation was carried out while using polystyrene as an exfoliating agent (ALDRICH). The polystyrene was dissolved in dichloromethane at a concentration of 3 mg/mL and powdered Molybdenum sulfide was mixed at a concentration of 1.5 mg/mL. The sample was sonicated while using a Sonic tip (FisherbrandTM Model 705 at 50% 700W for 5 ho per sample with a on/off sequence consisting in 1 s off and 1 s pulse and an ice bath was used to prevent solvent evaporation). After sonication, the dispersion was centrifuged at 1500 rpm for 45 min (Hettich Zentrifugen EBA 21(Hettich, Westphalia, Germany)).

The supernatant was preconcentrated and deposited on quartz films by rotation coating on 2 × 2 cm^2^ quartz substrate to prepare the films (APT-POLOS spin-coater(SPS-Europe B.V., Putten, The Netherlands): 4000 rpm, 30 s). The pyrolysis of polystyrene treatment was performed using an electric oven and using the following heating program: heating at 5 °C/min at 900 °C for 2 h.

For the preparation of films with random configuration, commercial MoS_2_ (Aldrich, St. Louis, Missouri, USA) at a concentration of 1.5 mg/mL and commercial graphene with an area of 700 m^2^ (STREM CHEMICALS, Newburyport, MA, USA) at a concentration of 7.5 mg/mL were added to a 30 mg/mL polystyrene solution (Aldrich, St. Louis, MO, USA) in dichloromethane and the suspension submitted to ultrasounds using a Sonic tip at 700 W for 5 h. After sonication, the dispersion was centrifuged at 1500 rpm for 46 min. and the supernatant used to prepare films of randomly configured MoS_2_/G heterojunction following the same procedure indicated above.

### 2.2. Methods-Catalytic Measurements

Electrochemical measurements were made in a conventional three electrode electrochemical cell while using a Pt disc pseudo-reference electrode and a Pt wire auxiliary electrode that accompanies the graphene modified glassy carbon (GCE) working electrode (BAS, MF 2012, area geometric 0.071 cm^2^). Voltammetric experiments were carried out with a potentiostatic device CH 920c (Cambria Scientific, Llwynhendy, Llanelli, Wales, UK), using H_2_SO_4_ 1.0 M saturated with air as an electrolyte. The working electrodes were the MoS_2_/fl-G films that were deposited on graphite. The graphite was deposited on the quartz films using the automatic carbon coater (sputter coater BALTEC_SCD005 (BAL-TEC AG, Schalksmühle, Germany) with carbon evaporation supplied BALTEC-CEA035).

## 3. Results

Upper and bottom layers of S atoms sandwiching and internal Mo IV layer constitute the crystal structure of MoS_2_. The geometrical arrangement of the three layers is such that top views define hexagons with alternating edges S and Mo atoms that are located at different planes. These hexagons are similar in size to those of G and, therefore, a superlattice configuration can be possible for MoS_2_-G heterojunctions when there is a coincidence of the two hexagonal arrangements. Figure 1 illustrates an ideal model of MoS_2_ structure and graphene layer assembled in the superlattice configuration. 

There are precedents in the literature showing the possibility to obtain MoS_2_/Graphene heterojunction with lattice matching geometry [25,26]. It was the leading hypothesis of the present study that the adaptation of this method could also serve to obtain MoS_2_/G superlattice since it has been previously reported the preparation of large area films of BN/G heterojunctions as superlattice [19]. 

Scheme 1 illustrates the procedure followed in the present study. As it can be seen there, the process starts with commercial MoS_2_ crystals that are subsequently exfoliated in viscous halogenated solvent while using polystyrene as promoter. After exfoliation, the residual bulk MoS_2_ crystals can be removed from the single and few layers MoS_2_ sheets by decanting the supernatant. Figure 2a presents AFM (Atomic Force Microscope) images of films of polystyrene containing MoS_2_. As expected in view of the plastic characteristic of polystyrene, these films were smooth and have a thickness about 200 nm (see profile in panel C of Figure 2). The presence of MoS_2_ in these films was not apparent from AFM at this moment due to the thickness. In a subsequent step, these films of polystyrene embedding MoS_2_ were submitted to pyrolysis in the absence of oxygen at 900 °C. These conditions have been previously reported to transform polystyrene into thin films of few layers defective of graphene [27]. One of the main advantages of the present procedure is its reproducibility and the possibility to prepare large surface areas.

A considerable shrinkage in the film thickness accompanies this transformation, as it can be seen in Figure 2b, where films of about 4 nm thickness corresponding to graphene can be seen (Figure 2d). The frontal image of MoS_2_/G film shows also the presence of MoS_2_ particles on this continuous graphene film. Measurement by AFM of a statistically relevant number of MoS_2_ particles on graphene indicate that the lateral size distribution ranges from 20 to 140 nm with an average about 80 nm and the thickness of the MoS_2_ is from 2 to 8 nm, with an average about 2. Appendix A illustrates the corresponding histograms of the lateral dimension and height. When considering that the interlayer distance of MoS_2_ is 0.615 nm, this average thickness corresponds to less than four MoS_2_ layers.

The SEM images of these films show a smooth surface with low apparent roughness and without cracks or pinholes. Figure 3a shows one of these representative SEM images. The TEM of these films were obtained by scratching a bit of these films. As an example, Figure 3b shows a representative TEM image of MoS_2_/G film after detachment from the quartz support.

The composition of MoS_2_ was confirmed by EDS elemental mapping, determining an overlap of Mo and S in 1:2 atomic ratio. Appendix A provides a summary of the Mo, S, O (from graphene defects), and C elemental distribution for a representative TEM image. These TEM images show a graphene layer larger than hundreds of nm having dark spots corresponding to smaller MoS_2_ particles of size from 20–140 nm. High resolution TEM images show the expected hexagonal arrangement characteristic of graphene and MoS_2_ (Figure 3c). Importantly, selected area electron diffraction at every point of the TEM image shows bright hexagonal spots corresponding to graphene. Coincident with those spots, there were also other points due to MoS_2_. The coincidence of the electron diffraction patterns between two 2D materials (Figure 3d) is taken as the best evidence for the superlattice configuration. 

It is similarly proposed here that the existing MoS_2_ sheets resulting from bulk MoS_2_ exfoliation are templating during the pyrolysis the formation of the nascent graphene layers in such a way that the growing graphene is replicating the existing MoS_2_ sheet resulting in lattice matching throughout the heterojunction, as in the precedent reporting the BN/G superlattice heterojunction. 

XRD and Raman spectroscopy convincingly evidence the presence of both components, graphene and MoS_2_, in the heterojunction. Appendix A shows the XRD pattern recorded for MoS_2_/G. This XRD shows a sharp peak at 2θ 14.60 corresponding to the 0.0.2 facet of MoS_2_. Other peaks of much lesser intensity correspond to other MoS_2_ planes. In addition to the peaks of MoS_2_, a broad diffraction band at 24° due to few layer defective graphene is also recorded [12].

Figure 4 shows the corresponding Raman for the MoS_2_/G heterojunction. At high wavenumbers, the expected broad peaks at 2840, 1540, and 1355 cm^−^^1^ corresponding to the 2D, G, and D peaks characteristic of defective graphenes are recorded. Two very sharp lines at 383 and 408 cm^−^^1^ that are characteristic of MoS_2_ are also seen. No changes in the position of the MoS_2_ or G bands is observed, indicating that the superlattice configuration does not alter the lattice constants of MoS_2_ or G, in agreement with the lattice match. The difference in wavenumber of these two peaks indicates the single layer (18 cm**^−1^**) or few layers structure of MoS_2_ (25 cm^−^^1^). In the present case, the difference between the A_1g_ and E_2g_ vibration modes of MoS_2_ was 24 cm^−^^1^, which agrees with the case of few layers MoS_2_. This configuration of MoS_2_ as few layers platelets is also in accordance with the previously commented AFM measurements of MoS_2_ particles that were supported on graphene.

### Catalytic Activity

The main purpose of this study is to determine the possibility of preparing large area films of superlattice MoS_2_/graphene (s-MoS_2_/G, s- meaning superlattice and corresponding to the material prepared, according to Scheme 1) heterojunctions that are suitable for electrocatalytic characterization and compare their electrochemical properties with those of a random MoS_2_/graphene (r-MoS_2_/G, r-meaning random) heterojunction, as stated in the introduction. Aimed at this purpose, an additional sample of r-MoS_2_/G was prepared by introducing commercial preformed graphene during the process of exfoliation of bulk commercially available MoS_2_ (Aldrich). Appendix A provides characterization data of r-MoS_2_/G sample used for the comparison, including a set of TEM images and EDS analysis with the corresponding elemental mapping. (Appendix A) Importantly, selected area diffraction patterns show the spots corresponding to hexagonal patterns of MoS_2_ and graphene not overlapped and clearly distinguishable for each 2D material, as expected for samples lacking the superlattice configuration (Appendix A, frame d). 

Electrochemical measurements were performed with a conventional three-electrode single cell with Pt as auxiliary electrode accompanying the MoS_2_/G electrode and a reference electrode. The quartz plates used for preparation of electrodes were previously modified by subliming carbon before MoS_2_/G film preparation, thus improving the electrical contacts, in order to increase the electrical conductivity of MoS_2_/G electrodes. Prior controls with sublimed carbon quartz plates lacking MoS_2_/G electrocatalyst did not exhibit any peak in cyclic voltammetry. Experiments were carried out for air-saturated, 1.0 M aqueous H_2_SO_4_ solution as electrolyte. Figure 5 shows the cyclic voltammetry corresponding to s-MoS_2_/G and r-MoS_2_/G under these conditions n. As it can be seen there, for s-MoS_2_/G a cathodic peak at −0.8 V vs. Ag/AgCl corresponding to ORR is recorded before a step peak corresponding to HER with onset being determined by extrapolation of the current density plot at −0.70 V. In the anodic region, the s-MoS_2_/G electrode exhibits the expected OER process with an onset potential of 0.87 V. For comparison, Table 1 compiles reported electrochemical data [28,29] for Pt/C and MoS_2_ electrodes together with the new data herein obtained for s-MoS_2_/G.

## 4. Conclusions

In summary, it has been shown that the exfoliation of bulk MoS_2_ crystals using polystyrene as promoter, followed by film casting and pyrolysis is a suitable procedure for obtaining few layers MoS_2_ deposited on few layers defective graphene in superlattice configuration. The procedure allows for obtaining large area films that are suitable for electrochemical characterization. By comparing the performance of two MoS_2_/G electrodes, one with lattice matching and another with random configuration in the heterojunction, it has been observed that the electrocatalytic activity of MoS_2_/G improves significantly in the superlattice configuration. This probably reflects the favourable orbital overlapping and electron migration in the MoS_2_/G heterojunction when the two lattices match one on top of the other. The present results show the far-reaching potential of superlattice assembly of 2D heterojunctions for application in electrocatalysis.

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
