# Peer review of "Superior Electrocatalytic Activity of MoS2-Graphene as Superlattice"

_nanomaterials, 2020, doi:10.3390/nano10050839_

Round 1

Reviewer 1 Report

The authors present a method for making MoS2-graphene in a superlattice structure. They characterize the structure and catalytic properties. This research can be helpful in finding non-precious metal catalysts for fuel cells and electrolysis.

The research and its presentation are generally good, however, a few details should be improved:

All Figures are blurred. Please use High-resolution files.

l 14: precise data to support the claim of 0.34V improvement are missing

l 20: a review on noble metal free catalysts should be included in the litrature

l 30-37: This refers to earlier work of the authors. This should be stated in the text.

l 64: the sentence does not make sense: ...1 s with 1 s...

l 125-129: please comment on oxygen distribution in Fig. S.2

l 149: 240  do you mean 24°?

l 154-161: what are the precise lattice constants of pure graphene and pure MoS2? Do you observe a distortion in the superlattice structure? Does this lead to a change in Raman frequencies?

l 170-172: what is what in the selective area diffraction pattern, Figure S.5d?

Figure 5: please use the english spelling of "potential"

l 189-190: please provide detailed onset-potentials for both catalysts. You will need a second decimal to justify the 0.34 V claimed in line 14.

l 1, 202-203: Will the highlited information be filled in by the authors or by the publisher?

Author Response

The present manuscript describes the preparation and characterisation of MoS2/graphene superlattice electrocatalyst for HER, OER and ORR. The manuscript is well written and all the experimental results are convincing. I have only minor remarks.

General question: How reproducible the results are?

The most clear advantage of the procedure reported in the manuscript is reproducibility that derives from the reliability of the pyrolysis. A comment on this has been introduced in the revision in page 3 lines 111-113.

Page 2, line 63: It must be „powdered“ Molybdenum sulfide.

Mispelling has been corrected.

Page 4, line 112: It must be MoS2.

The missing letter was typed.

Page 6, line 167: What is commercial preformed graphene, where does it come from, or how it was prepared?

The starting material was purchased from Aldrich. This has now been indicated. (page 6, line 180).

Reviewer 2 Report

The present manuscript describes the preparation and characterisation of MoS2/graphene superlattice electrocatalyst for HER, OER and ORR. The manuscript is well written and all the experimental results are convincing. I have only minor remarks.

General question: How reproducible the results are?

Page 2, line 63: It must be „powdered“ Molybdenum sulfide.

Page 4, line 112: It must be MoS2.

Page 6, line 167: What is commercial preformed graphene, where does it come from, or how it was prepared?

Author Response

This paper demonstrates that the preparation of hybridized materials of graphene and MoS2. The authors study their structures using AFM, TEM, SEM, and Raman, as well as electrocatalytic performances. I feel this paper has potential interests; however, I recommend the publication after addressing the following issues.

1)There are no experimental details on how to make s-MoS2/G and r-MoS2/G, as well as how to make inks for the measurement of electrocatalytic performances.

s-MoS2/G was obtained as indicated in scheme 1 and commented in the text. r-MoS2/G was obtained by mixing exfoliated MoS2 and preformed G. This is now indicated in the revised version in page 2 lines 72-78. No inks were used since the films were directly used as electrodes.

2) Need to show the structural difference between s-MoS2/G and r-MoS2/G.

The difference in the selected area electron diffraction pattern showing the coincidence (superlattice, s-MoS2/G) or not (r-MoS2/G) of the MoS2 and G lattices can be inferred by comparison of Figure 5d with that of the Figure S5 in the supporting information.

3) Need to show electrocatalytic performances comparing with graphene, MoS2, and standard Pt/carbon.

Following the recommendation of the reviewer, HER data for reference materials, Pt/C, and MoS2, have been added and compiled in new Table 1.

4) There is no structural evidence that used materials are graphenes.

Formation of graphene from polystyrene and the corresponding full characterization have been reported as indicated in the text. (page 3, line 109-111)

5)This paper needs the characterization of diffraction spots of s-MoS2/G compared with r-MoS2/G.

As indicated in previous point 2, the required selected area electron diffraction pattern of s- and r- MoS2/G serves to determine the superlattice or random configuration of MoS2/G lattices. Other characterization data of r-MoS2/G are provided in Figures S4 and S6.

Reviewer 3 Report

This paper demonstrates that the preparation of hybridized materials of graphene and MoS2. The authors study their structures using AFM, TEM, SEM, and Raman, as well as electrocatalytic performances. I feel this paper has potential interests; however, I recommend the publication after addressing the following issues.

1)There are no experimental details on how to make s-MoS2/G and r-MoS2/G, as well as how to make inks for the measurement of electrocatalytic performances.

2) Need to show the structural difference between s-MoS2/G and r-MoS2/G.

3) Need to show electrocatalytic performances comparing with graphene, MoS2, and standard Pt/carbon.

4) There is no structural evidence that used materials are graphenes.

5)This paper needs the characterization of diffraction spots of s-MoS2/G compared with r-MoS2/G.

Author Response

(The authors gave the same response as above.)

Round 2

Reviewer 3 Report

This paper was revised and improved along with the reviewers' comments. I recommend the publication after correcting about how much the percentage of platinum in carbon used as the reference sample.

Author Response

The Pt content was 20 %.

The manuscript has been modified adding: Pt/C (Johnson-Matthey, 20 wt% Pt/XC-72) and  the change highlighted in green.